# Dynamic Multi-Axis Calibration of MEMS Accelerometers for Sensitivity and Linearity Assessment

**DOI:** 10.3390/s25072120

**Published:** 2025-03-27

**Authors:** Luciano Chiominto, Giulio D’Emilia, Antonella Gaspari, Emanuela Natale

**Affiliations:** 1Department of Industrial and Information Engineering and Economics, University of L’Aquila, 67100 L’Aquila, Italy; giulio.demilia@univaq.it (G.D.); emanuela.natale@univaq.it (E.N.); 2Department of Mechanics Mathematics and Management, Polytechnic of Bari, 70125 Bari, Italy; antonella.gaspari@poliba.it

**Keywords:** three-axis micro-electromechanical systems (MEMS) accelerometer, multi-axis calibration, uncertainty modeling, uncertainty assessment

## Abstract

A set of commercial triaxial micro-electromechanical systems (MEMS) accelerometers was calibrated using a custom-designed test bench featuring a rotating table. The calibration setup enabled simultaneous assessment of all accelerometer measurement components, generating precise reference accelerations within a frequency range of 0 to 8 Hz. A working model of the calibration setup and procedure was described to provide a complete uncertainty budget for both the reference and sensor accelerations. Through experimental uncertainty assessment of all the accelerometers, linearity and sensitivity were evaluated at different sensor levels. These parameters were determined by considering a single value for each accelerometer and detailing the analysis for each axis. Data processing revealed the achievable level of uncertainty and how it was influenced by the evaluation method employed for analyzing the calibration data.

## 1. Introduction

Micro-electromechanical systems (MEMS) technology is widely acknowledged for its ability to enable sensors useful for many applications and in different engineering sectors. For example, MEMS accelerometers are employed in aerospace, civil, automotive and automation fields. These devices have many interesting measurement characteristics, such as miniaturization, high metrological performance, digital output for easy networking, and low cost [1]. MEMS accelerometers have gained a relevant role thanks to the possibility of realizing triaxial accelerometers for different applications. Their cost effectiveness makes them ideal for large-scale deployment in structural health monitoring (SHM) applications, where many sensors are required across vast areas (e.g., bridges, entire buildings, or groups of buildings).

The ability to deploy many sensors increases the likelihood of detecting localized issues, providing a more comprehensive picture of the overall health of a structure [2,3,4]. In this field, low-frequency range vibrations (typically below 100 Hz) are of particular interest since they are often associated with the natural frequencies of structures [5].

As the number of potential applications continues to grow, so does the demand for meeting metrological requirements, including reproducibility and traceability. In this regard, some problems arise when metrologically characterizing these sensors, mainly due to the low-cost production processes and their production volumes. Furthermore, defining a low-cost calibration procedure for digital sensors is also a difficult task since many causes of uncertainty should be considered. For example, zero-bias, sensitivity nonlinearity, cross-axis effect related to non-orthogonality, and the crosstalk effect between different channels caused by the sensor electronics, temperature and environmental effects [6,7]. Moreover, modeling all these causes of error is not trivial, nor is it easy to reliably estimate the uncertainty. For this reason, different calibration procedures have been developed that could be used in practical cases [6,8].

As proposed by different authors, the calibration techniques for MEMS accelerometers can be generally divided into:Non-autonomous (classical) calibration methods: these are realized by means of precise devices with very accurate mechanical structures, such as a high-precision turntable, centrifuge, shaking table, etc.Autonomous calibration methods: these are completed by using an external reference excitation without the assistance of high-precision instruments. This can be provided by the local gravity field, the rotational angular velocity of the Earth, a uniform magnetic field, etc.

Both classes of calibration methods present advantages and disadvantages. In non-autonomous (classical) calibrations, the main requirement is that an accurate acceleration standard has to be realized. This is obtained by checking the geometrical and motion characteristics. Another important aspect is the relative positioning of the device under test with respect to the calibration bench since misalignments are a further source of uncertainty [9,10]. Several different test benches have been realized, which have made significant contributions to static and dynamic calibration [11,12,13,14,15,16,17,18].

In autonomous calibration methods, gravitational acceleration is typically used as a reference value. The calibration is performed by changing the orientation of the measuring axes of the accelerometer with respect thereto. Then, the primary sensitivities, cross-axis sensitivities, and bias values are obtained by performing a data-fitting process based on different positions. In this case, particular attention should be given to the geometrical configuration of the accelerometers’ axes, along with the angle between them. These are not exactly right angles.

In general, a matrix method is used to model geometrical errors [6,19], and different algorithms have been developed to improve the fitting of the data. These include the Allan variance [19], the Kalman filter [20], and the ellipsoid method [21].

In some cases, the reference angles relative to the vertical direction are achieved by embedding the triaxial accelerometer in a multifaced device. This device has multiple positions that are inclined at different angles to the vertical, all without the need for a turntable or angular encoders [22]. Of course, the geometrical accuracy of the device used to realize angles of inclination of the reference device affects the measurement uncertainty and the bias estimation. However, very accurate results could be achieved without any prior assumption [9]. In other cases, the matrix method was developed to make the method of computation of primary sensitivities and bias more robust by identifying outliers [23].

A merging approach based on a rotating device without using it as an acceleration reference is proposed in [24]. In this case, a tilted wheel is mounted on a static support. The accelerometer is fastened to the bench, and then the wheel is manually set into rotation. The acceleration signal is acquired while the bench is freely rotating. The proposed calibration procedure estimates both the calibration parameters and the bench motion with no prior knowledge about the bench rotation. Only the distance of the sensor from the rotation axis needs to be known, which can be easily obtained through direct measurement on the bench. The estimation accuracy of the sensitivity values depends on the noise level, typically being less than 1%, as a percentage value.

Environmental effects are also studied, in particular, drift effects due to temperature changes [25]. Nevertheless, they are beyond this analysis since it focuses on calibration and traceability aspects typical of a laboratory calibration.

Bearing in mind the previous considerations, many opportunities arise from autonomous approaches. However, generally, only the geometrical aspects are studied and evaluated. These have an important effect but are not the whole contribution to the uncertainty budget. In [6], an interesting and detailed comparison of many calibration methods was carried out, although it was qualitative. Autonomous approaches can be realized in different ways and offer the possibility of reducing the cost of calibration. However, only geometrical aspects and electronic cross-sensitivity effects could be corrected by these methods. Furthermore, managing all the required geometrical data makes it difficult to develop simple procedures, and complex uncertainty models arise. A possible simplification for uncertainty evaluation could come from a Bayesian approach for calibration [7], even though further validation for more complex scenarios is required.

An important effect to be considered during calibration is the linearity of MEMS transducers with reference to the whole measuring range. This becomes more significant when considering low-cost sensors. These are typically able to measure acceleration up to ±10 g; therefore, using input accelerations in the range of ±1 g makes it difficult to assess linearity, even though some efforts in this direction have been made [1,19]. Sensor linearity could be assessed only by using a non-autonomous approach.

Based on these aspects, in this work, a non-autonomous methodology for multi-axis MEMS accelerometer calibration was realized, based on a rotary test bench. The calibration test bench was designed and tested to perform multi-axis dynamic sensitivity analysis by alternative oscillations driven by an electrical motor with a high-accuracy angular encoder. It can calibrate in the same test two measuring axes simultaneously subjected to a rotating motion. The third axis, along the vertical direction, was subjected to a nominal constant acceleration equal to ±g. This output was used only for alignment purposes. In [26], the same rotating bench was compared with a classic linear slide, and the results were found to be compatible. In particular, the comparison was carried out axis by axis and for each sensor, highlighting comparable results, considering standard uncertainty levels of 1.0–1.5% for the rotating bench. It should be noted that improvements in the operating procedure and uncertainty evaluation led to a lower estimated uncertainty, which, in any case, does not change the statistical compatibility with the results previously obtained with the linear slide.

A detailed model of the calibration system is described to obtain a complete uncertainty budget of the estimated primary and cross-sensitivities and information about sensor linearity. Building on previous work by the authors [26,27,28,29], many MEMS accelerometers were examined with the aim of determining the uncertainty to be assumed. Carefully modeling the experimental and data processing effects allowed us to consider different cases to make a specific and novel contribution toward a conscious and reliable use of these accelerometers: evaluating the sensitivity (and its uncertainty) of an entire batch of accelerometers, studying individual sensors, or providing metrological specifications for each axis of each accelerometer. In Section 2, the methodology for constructing a suitable calibration uncertainty budget and the experimental program is described. In Section 3, the results are presented and discussed, and the Conclusions end the paper.

## 2. Materials and Methods

### 2.1. Devices Under Test

Eight commercial, low-power, digital, MEMS accelerometers by STMicroelectronics (STMicroelectronics, Agrate Brianza, Italy), belonging to the same production batch, were examined in this study. Each accelerometer was part of an inertial measurement unit (LSM6DSR model [30]) composed of an analog-to-digital converter, a charge amplifier, a 3D accelerometer, and a 3D gyroscope. Since only sensitivity related to acceleration was tested for the purposes of this study, the 3D gyroscope was not used and kept off. A serial cable connected the digital MEMS accelerometers to an external microcontroller (STMicroelectronics, model 32F769IDISCOVERY [24]). Using a Universal Serial Bus (USB) cable, the microcontroller was connected to a PC for transferring the digital data acquired by the sensor. Amplitude values ranged between −2^16 − 1^ = –32,768 D_16 bit-signed_ and +(2^16 − 1^ − 1) = +32,767 D_16 bit-signed_, where the digit unit is a signed 16-bit sequence converted into a decimal number. The sensitivity of the digital MEMS accelerometers is expressed by the manufacturer in terms of mg/LSB, where g is the gravitational acceleration and LSB means least significant bit. Sensitivity also depends on the set “full scale.” With a “full scale” of ±8 g, the sensitivity declared is 0.244 mg/LSB [31], corresponding to 418 D_16 bit-signed_/(m/s^2^).

### 2.2. Calibration Bench

The MEMS under test were calibrated using a polylactic acid (PLA) 3D-printed disk. It rotated around its symmetry axis, and was mounted on the axis of a Schneider Electric (Rueil-Malmaison, Hauts-de-Seine, France) servomotor. The calibration bench is presented in Figure 1.

To study all the axes, the MEMS sensors were placed on different 3D-printed supports. These were specifically made to change the MEMS orientation with respect to the tangential and centripetal accelerations, so that, in each configuration, two axes were simultaneously stressed. Three supports were made to achieve the following positioning:*x*-axis parallel to the tangential acceleration, *y*-axis—to the centripetal acceleration (Figure 2a);*z*-axis parallel to the tangential acceleration, *x*-axis—to the centripetal acceleration (Figure 2b);*z*-axis parallel to the centripetal acceleration, *y*-axis—to the tangential acceleration (Figure 2c).

Another support is used for attaching the microcontroller to the disk.

### 2.3. Reference Acceleration

In this paper, the reference acceleration values for calibration were determined based on the angular position of the servomotor axis, provided by a high-accuracy encoder.

The programmable logic controller (PLC) that allows the management of the rotating disk provides the value of the angular position of the servomotor axis over time:(1)$$\theta \left(t\right)=\theta_{0}\cos\left(2\pi ft\right)$$

In addition, the system also provides the angular speed $\omega \left(t\right)$, obtained as the derivative of $\theta \left(t\right)$:(2)$$\omega \left(t\right)=\dot{\theta }\left(t\right)=-2\pi f\theta_{0}\sin\left(2\pi ft\right)=\omega_{0}\sin\left(2\pi ft\right)$$
where $\omega_{0}=-2\pi f\theta_{0}$.

This information was used to calculate the reference centripetal and tangential accelerations, as described in the following:(3)$$a_{r}=r{\dot{\theta }}^{2}\left(t\right)=r\omega^{2}\left(t\right)=r{\left(\omega_{0}\sin\left(2\pi ft\right)\right)}^{2}=r\frac{\omega_{0}^{2}}{2}\left(1-\cos\left(2\pi 2ft\right)\right)$$(4)$$a_{t}=r\ddot{\theta }(t)=r\dot{\omega }(t)=2\pi fr\omega \left(t\right)=2\pi fr\omega_{0}\cos\left(2\pi ft\right)$$
where *f* is the oscillation frequency and r is the distance of the sensitive element of the accelerometer under test from the rotation axis.

The supports were designed and installed on the disk in such a way that the center of the sensor under test (where the sensitive element was placed) was positioned at a nominal distance from the axis of rotation equal to r = 170 mm (standard uncertainty equal to 0.5 mm, or 0.3%).

It should be noted that the centripetal acceleration, as it appears in Equation (3), is characterized by double the frequency with respect to the tangential one. This aspect must be taken into consideration and represents the main difference between the curvilinear calibration bench and linear ones, where the axes are stressed by accelerations at the same frequency. It should also be noted that the constant term in the radial acceleration is equal to the amplitude of the acceleration itself, which means it could be simply obtained by averaging the signal.

In addition, a second reference Integrated Electronics Piezo-Electric (IEPE) accelerometer was used for validation and comparison purposes, a PCB Piezotronics (Depew, NY, USA) model TLD356B18 with a measurement range of ±5 g. This accelerometer was positioned at a distance of 145 ± 0.1 mm from the rotation axis. It was used for considering other possible error sources, such as vibration on the rotating disk, and to check the dynamic characteristics of the motor.

### 2.4. Test Plan

The tests were carried out considering two different oscillation frequencies: 5 Hz and 8 Hz.

For each frequency, the oscillation angle was set so as not to exceed the limit value of ±5 g, which is the maximum acceleration that can be measured using an IEPE accelerometer. Therefore, positioning MEMS at a radius of 170 mm, the theoretical testing conditions were as follows:
5 Hz as the oscillation frequency and $\theta_{0}$ angle at 34°: the amplitude of the tangential acceleration was equal to 49 m/s^2^, and the centripetal acceleration was equal to 15 m/s^2^.8 Hz as the oscillation frequency and $\theta_{0}$ angle at 13°: the amplitude of the tangential acceleration was equal to 49 m/s^2^, and the centripetal acceleration was equal to 6 m/s^2^.For each experimental setup, the data from the digital MEMS, the IEPE accelerometer, and the motor encoder were recorded for 40 s. Each test was repeated 6 times.

### 2.5. Data Processing

#### Sensitivity Evaluation

The purpose of data processing was to evaluate the amplitude of the measured acceleration and the reference acceleration and to compute their ratio to determine the sensitivity of the tested accelerometer at the given oscillation frequency.

The signal from the digital MEMS and the IEPE accelerometer was filtered using a 6-pole digital low-pass Butterworth filter with a cutoff frequency of 40 Hz [27]. From the encoder data, the centripetal and tangential acceleration amplitudes were calculated using Equations (3) and (4). A zero-crossing algorithm was applied to all the signals (from the accelerometer and the encoder) to select complete periods. This procedure was implemented to reduce the leakage during the further analysis. All the data processing steps were made using MATLAB R2023b.

To determine the acceleration amplitude, the sampling frequency of the signal was necessary. As stated in previous works by the authors [27], the sampling frequency of digital MEMS is not constant over time. Moreover, the mean value appears not to be equal to the selected one, and it is not constant among different sensors. To evaluate the mean sampling frequency, the following procedure was developed.

The signal is processed using the fast Fourier transform (FFT) at different sampling frequencies ranging from 1560 Hz to 1760 Hz with a step of 1 Hz.

Then, the mean sampling frequency is identified as the one that maximizes the amplitude at the oscillation frequency (5 Hz or 8 Hz).

For the tangential acceleration, the harmonic of interest corresponds to the frequency of the oscillation itself. In contrast, the centripetal acceleration theoretically exhibits two significant harmonic components: the first is at twice the frequency of the oscillation, and the second is a constant component, representing the steady-state part of the centripetal acceleration (refer to Equation (3)). Figure 3 represents the pseudocode of the developed sampling frequency variability compensation method.

The algorithm is based on the following main routines:Initialization: An array of frequencies is defined that ranges from TheoricalSamplingFrequency − 100 to TheoricalSamplingFrequency + 100, and an array is created to store the errors.FFT calculation: for each frequency in the range, the FFT of the input data is calculated, and the spectral amplitude is obtained.Finding the maximum amplitude and frequency: the maximum amplitude and the corresponding frequency are identified.Error calculation: the error between the identified maximum frequency and the target is calculated.Finding the real frequency: the index of the minimum error is found, and the corresponding frequency is returned.

The contribution to the variability of this procedure should be considered as part of the measurement repeatability.

Finally, for each test, the sensitivity of the tangential axis and the centripetal axis were calculated.

The sensitivity values obtained are finally averaged across different groups to investigate the possibility of assuming:A single sensitivity value for all accelerometers. This is supposed to be independent of the accelerometer, the measuring axis, and the way the sensitivity is evaluated. This is the most general and easy approach for in-field use. However, it is important to estimate the uncertainty of the method.A single value of sensitivity for each accelerometer, common for all three measuring axes.Specialization of the sensitivity indication to the single axis of a single accelerometer. This approach seems to be the most accurate one, even though a significant amount of data needs to be acquired before the accelerometer can be used.

### 2.6. Uncertainty Evaluation

For the evaluation of uncertainty of the sensitivity, repeatability, linearity, and uncertainty of the reference contributions were considered.

#### 2.6.1. Repeatability

The repeatability contribution to uncertainty was assessed by performing six repeated measurements for each experimental setup. The sensitivity was calculated for each test, and the repeatability contribution ($u_{r}$, expressed in percentage terms) was evaluated as the standard deviation of the calculated sensitivities, according to a Gaussian probability distribution.

#### 2.6.2. Linearity

Linearity was assessed by considering the output data of each sensor axis under different acceleration stresses. In fact, as previously mentioned, during the tests, the acceleration amplitudes ranged from the lowest value of 4 m/s^2^ to a maximum of about 50 m/s^2^, which are of interest for different applications in the automotive sector, e.g., airbag deployment, advanced driving assistance systems, or navigation and positioning.

To assess the linearity uncertainty, the data regression straight line to model the relationship between the MEMS output value and the encoder reference accelerations was calculated using the least squares error method:(5)$$q_{o}=Sq_{i}+b$$
where $q_{o}$ is the output value of the MEMS accelerometer [D_16-bit-signed_], $q_{i}$ is the reference acceleration [m/s^2^], b is the intercept, and S is the slope of the calibration line, which corresponds to the sensitivity of the dynamically linear instrument [D_16-bit-signed_/(m/s^2^)]. According to [32], the standard deviation $\sigma_{S}$ of the sensitivity can be calculated as follows:(6)$$\sigma_{S}=\sqrt{\frac{N\sigma_{q_{o}}^{2}}{N\sum q_{i}^{2}-{\left(\sum q_{i}\right)}^{2}}}$$
where:(7)$$\sigma_{q_{o}}^{2}=\frac{\sum {\left(Sq_{i}+b-q_{o}\right)}^{2}}{N-2}$$

Then, $u_{l}=\frac{\sigma_{S}}{S}\cdot 100$ can be considered as the linearity uncertainty contribution of the sensitivity S, expressed in percentage terms.

To express the goodness of fit for the linear regression model, the coefficient of determination R^2^ was used.

#### 2.6.3. Uncertainty of the Reference

The uncertainty of the reference depended on the uncertainty of the encoder, which was very low (0.01°).

To estimate the quality of the reference, the following preliminary evaluations were carried out:Agreement between the performed movement and a theoretical sinusoidal motion.Calculation of the repeatability of the measured amplitude of the reference acceleration in repeated tests.Verification of consistency of the amplitude of the angular position signal with the set oscillation angle and of the fact that the amplitude of the angular velocity signal was the position signal’s amplitude multiplied by the pulsation, as in perfectly sinusoidal signals.Comparison with the IEPE accelerometer described in Section 2.3.

The agreement between the obtained movement from the servomotor and a sinusoidal one was obtained by calculating the FFT of the angular signal from the encoder. The amplitude at the oscillating frequency was used to calculate the maximum angular velocity in a perfect sinusoidal movement. This value was compared to the maximum angular velocity value obtained by the encoder.

However, the reference primary source of uncertainty was represented by geometrical imperfections of the test bench. These were due to the inclination of the rotation axis with respect to the vertical axis, and to the deviations in the position of the MEMS accelerometer under test with respect to the theoretical position. The MEMS sensing element was positioned at a distance of r = 170 mm ± 0.3% from the rotation axis. This was measured by considering the sensing element located at the geometrical center of the MEMS. Equations (3) and (4) were used for calculating the reference accelerations. Both assume a perfect positioning of the MEMS, which is not feasible.

An analytical model was built to evaluate the reference uncertainty contribution due to the geometrical imperfections of the bench. The parameters used in the model are shown in Figure 4 and described in Table 1. The term “reference side” in Figure 4 indicates the direction from which angle δ is measured clockwise and corresponds to the direction of the maximum slope of the disk.

As described in Section 2.2, the experimental tests were conducted by positioning each MEMS accelerometer in such a way that the measurement axes were oriented both in the radial and tangential directions. As an example, in Figure 4b, the *x*-axis is represented in the radial direction and the *y*-axis is in the tangential direction.

It must be considered that the tangential displacement Δs produces a variation in the radius at which the sensitive element is placed, as well as a rotation of the accelerometer around its own axis. For this reason, at Step 2 of the procedure, r and β were redefined as indicated in the diagram itself (See Figure 5).

The α angle was preliminarily tested using a triangulation laser aimed near the edge of the disk at regular distances along the entire circumference: these tests highlighted a tilt angle well below 1°. Further checks using a precision level confirmed these data.

The model was applied assuming a maximum value of 1° for α, of 0.5° for β, and of 1 mm for Δs. This choice, in the tests at 5 Hz, involved a 5 Hz component in the radial direction not exceeding 0.15 m/s^2^: in the experimental tests, in fact, the correct positioning of the accelerometer was always checked, precisely by verifying that this component did not exceed 0.15 m/s^2^. This verification allowed keeping positioning imperfections within the specified limits.

In fact, negligible deviations (less than or equal to 0.06%) occurred for the components at 5 and 10 Hz in the radial and tangential directions, while the largest deviations from the ideal case (test bench without geometrical imperfections) were found to be of the order of 1.2% for the constant acceleration component of the axis in the radial direction. This result led us to exclude the calculation of sensitivity using data for a null frequency. The geometrical contribution to the uncertainty evaluation was a maximum error in the order of 0.06%, excluding radius-related uncertainty.

## 3. Results

The results of the experimental activity were examined to estimate the following elements:Metrological characteristics of the MEMS accelerometers;Differences between the accelerometers, to check the possibility of using different data depending on the behavior of the individual sensor;Differences between the indicators.

### 3.1. Metrological Characteristics

The metrological characteristics of MEMS accelerometers are repeatability, linearity, and sensitivity. To evaluate the quality of the experimental data, repeatability was the first indicator calculated. Then, linearity and sensitivity analysis were assessed for each axial component of all the accelerometers. The total uncertainty of the sensitivity was the combination of the repeatability and linearity of the MEMS along with the uncertainty due to geometrical imperfections of the bench and the uncertainty of the reference.

#### 3.1.1. Repeatability

For each experimental setup, the acquisitions were repeated six times to evaluate the repeatability of the sensitivity measurements. This was calculated for each accelerometer axis stressed by the accelerations at different frequencies and amplitudes. By analyzing the test data, the mean repeatability was in the order of 0.05%. This result also provided an indication of the quality of the procedure used for the compensation of the sampling frequency variability.

#### 3.1.2. Linearity

To assess the sensors’ linearity, data of both the tangential and centripetal accelerations were used. An example of the linearity of all the axes of a sensor is shown in Figure 6. There were no significant differences in linearity between the three axes in the measuring range. In addition to this, no noticeable biases were found.

The coefficient of determination R^2^ was more than 0.99 for all the axes, demonstrating a strong relation between the linear model and the experimental data. The correct fitting of the linear model was also assessed by considering the residual plots. As can be seen in Figure 6, the residuals presented a random pattern, showing no systematic deviation from the horizontal axis. This suggests that the regression model adequately captured the variability in the data.

According to the procedure described in Section 2.6.1, the calculation of $u_{l}$ led to values less than 0.05%.

#### 3.1.3. Uncertainty of the Reference

The repeated tests carried out on the acceleration reference, according to Section 2.4, highlight an almost perfect repeatability (very negligible standard deviation).

Considering the generated movement, this appeared to be perfectly sinusoidal since in the FFT of the angular position, the only frequency recognized was the one at the oscillation. Furthermore, using the maximum amplitude of the position to calculate the maximum angular velocity showed an agreement between these two values. This was reflected in differences less than 1‰ in terms of the instantaneous amplitude of acceleration.

Reference uncertainties due to the imperfections of the bench were evaluated according to the procedure described in Section 2.4. Considering an oscillation frequency of 5 Hz, deviations of less than or equal to 0.06% occurred for the components at 5 and 10 Hz, in the tangential and radial directions, respectively. If the rectangular distribution is considered, the maximum uncertainty contribution, as a standard deviation, is $u_{g}=0.03\%$ ($\frac{0.06\%}{\sqrt{3}}$), which is negligible.

The percentage uncertainty of the radius, estimated to be on the order of 0.3%, was reflected as such in the reference acceleration values, since both the radial and tangential accelerations were proportional to the radius. Therefore, considering also the uncertainty of the radius $u_{r}^{2}$ = 0.3%, the uncertainty of the reference was estimated as follows:(8)$$u_{ref}=\sqrt{u_{g}^{2}+u_{r}^{2}}=\sqrt{0.03^{2}+0.3^{2}}=0.3\%$$

For validation purposes, the acceleration data from the encoder and the data provided by the IEPE sensor were found to be statistically compatible [28].

Various statistical techniques can be used to check the agreement between different measurement methods, as mentioned in [33]. These include ANOVA, regression, error analysis, standard deviation, and intraclass correlation coefficient (ICC), some of which were considered. An additional specific validation action for the agreement of the two sets of acceleration data is presented in the Bland–Altman plot in Figure 7.

The Bland–Altman plot demonstrates the agreement of the acceleration data provided by the encoder and the piezo IEPE accelerometer. A negligible bias in the difference is acknowledged, and it is equal to −0.08 m/s^2^. The calibration bench uncertainty is within the agreement limits.

#### 3.1.4. Budget for the Calibration Uncertainty Assessment

The overall uncertainty of the sensitivity, evaluated using the described rotating calibration bench, was obtained by combining all the abovedescribed contributions. Table 2 shows the uncertainty budget.

For a conservative analysis, the maximum possible correlation (correlation coefficient equal to one) between the different pairs of contributions was considered.

### 3.2. Sensitivity Obtained Across Groups of Data, and the Corresponding Uncertainty

As mentioned in Section 2, the sensitivity values obtained in the test of each accelerometer were averaged across different groups of data. The aim was to investigate the possibility of assuming, for all the accelerometers of a batch, a single sensitivity value and determining how this affects uncertainty with respect to each sensor’s calibration. This latter approach would be the most accurate, even though it is not feasible in practice.

This analysis required the following steps of evaluation:Assessment of the mean sensitivity for each axis and each accelerometer.Assessment of the mean sensitivity for each accelerometer.Assessment of the mean sensitivity across all the axes and accelerometers.

#### 3.2.1. Assessment of the Mean Sensitivity for Each Axis and Accelerometer

Figure 8 shows the sensitivity values obtained as the average for each axis and each accelerometer. The uncertainty, represented by “error bars,” was obtained by combining the calibration uncertainty (0.4%) with the standard deviation of the sensitivity results obtained for each axis and each accelerometer: on average, the overall uncertainty was equal to 0.8%, 0.6%, and 0.7%, for the *x*-, *y*-, and *z*-axes, respectively.

Using the same calibration method, Figure 6 shows that the sensitivity values, as well as the variability, were similar for all the sensors. However, MEMS 3 showed higher sensitivity values compared to the other systems, while MEMS 1 had a greater variability on the *x*-axis. These experimental results were due to the characteristics and inconsistencies of the individual sensors.

#### 3.2.2. Assessment of the Mean Sensitivity for Each Accelerometer

Figure 9 presents the sensitivity values for each accelerometer independently of the axis. The sensitivity of each accelerometer was computed as the mean value with respect to all the axes. The sensitivity values ranged from 421.5 to 426.6 D_16 bit-signed_/(m/s^2^). The uncertainty was obtained as a combination of the variability of the first case and the variability among the axes: on average, the overall uncertainty was 0.9%.

#### 3.2.3. Assessment of the Mean Sensitivity Across All the Axes and Accelerometers

Finally, considering a single mean sensitivity value for all the accelerometers, it was estimated to be equal to 423.0 D_16 bit-signed_/(m/s^2^) (Figure 10). In that case, the uncertainty was obtained as a combination of the variability of the previous case and the variability among the MEMS, and it was equal to 1%.

The obtained sensitivity values were compared to assess compatibility between the MEMS. This was performed using an analysis of variance (ANOVA) and the normalized error. Considering the ANOVA, the sensitivity values across the three axes of each MEMS were compared. In that case, all the accelerometers were compatible with each other.

The normalized error was calculated using a single sensitivity value for each MEMS and the corresponding uncertainty. Most accelerometers were compatible with each other, providing normalized error values less than 1. Only MEMS 3 was not compatible with MEMS 6. This evidence suggests that in order to consider all the accelerometers compatible, the estimated uncertainty has to be increased.

It should be noted that assigning a single value is the approach followed by the manufacturer. In fact, in the sensor datasheet [30], the provided sensitivity value at component level is 418 D_16 bit-signed_/(m/s^2^). However, this does not include an uncertainty budget nor a traceability statement.

Table 3 synthesizes these overall uncertainty estimates. If a single sensitivity value for all the accelerometers of the batch is provided, determined on a single axis of a single accelerometer, then the uncertainty of Case 3 should be considered. As expected, it was higher than in the other cases.

### 3.3. Sensitivity Obtained Across Groups of Data, and the Corresponding Uncertainty

The estimation of each axial sensitivity was carried out by making measurements of both the tangential and centripetal acceleration, but also by considering vertical data. These were related to the measurement of gravitational acceleration.

Tangential measurements are, in general, affected by less variability than radial ones when both mean and periodic values are considered (at two times the oscillation frequency). This evidence could be explained by considering the acceleration amplitudes. Tangential accelerations have higher amplitudes compared to radial ones therefore, the signal-to-noise ratio is better. In addition to this, the misalignment effects of the test bench are reduced. As far as the constant component of the radial acceleration is concerned, it can also be used to evaluate the sensitivity of accelerometers. Nonetheless, it is more influenced by geometrical imperfections of the oscillating disk, and constant offsets are present.

It is important to highlight that the evaluated sensitivity of all the axes of all the accelerometers was higher than the nominal one. This result suggests that a promising way to use the calibration data is to correct the sensitivity data according to the calibration results, using a common value for the sensitivity itself for all the accelerometers, with an extended uncertainty of about 2% (at 95% confidence level), provided that the level of uncertainty, the calibration uncertainty contribution, and the effect of positioning of the transducer are acceptable. Should the actual sensitivity deviate systematically from the nominal value provided by the manufacturer, it will inevitably introduce a bias into the final measurement results—an eventuality that demands careful consideration in many in-field applications exploiting the use of MEMS accelerometers.

The average for each accelerometer axis allows for a reduction in variability and can be evaluated if a lower level of uncertainty is required.

## 4. Conclusions

In this work, a rotary test bench for three-axis MEMS accelerometer calibration was used to carry out a dynamic sensitivity analysis of a set of sensors from the same production batch. The main results can be synthesized as follows:Differences between the accelerometers and between the axes of the same accelerometer were not significant from a statistical point of view, considering the variability.Tangential measurements are, in general, affected by less variability than radial ones due to the higher amplitudes.The constant component of the radial acceleration could be used to evaluate the sensitivity of the accelerometers. However, it was influenced more by geometrical imperfections of the bench, and constant offsets were present.The evaluated sensitivity of all the accelerometers’ axes was greater than the nominal one.

In light of these results, a possible practical approach to the problem could be to first evaluate the sensitivity using a limited set of accelerometers from the same batch. Secondly, to analyze them on the proposed calibration bench. Finally, to consider a single value for the sensitivity itself for all the accelerometers with an extended uncertainty of about 2% (at 95% confidence level). If a lower level of uncertainty is required, analyzing each accelerometer and each axis reduces variability.

In applications that require high-precision measurements, such as in aerospace and precision engineering, the uncertainty of the proposed calibration method may not meet the requirements. On the other hand, in these cases, stringent requirements and environmental criticalities necessitate the use of more accurate and reliable types of sensors. However, the applicability of the method is very broad in various fields of use, where the requirement of low cost of both the sensor and the calibration method is a priority: the evaluations carried out in this work can help to address the choices appropriately. As a final remark, it is important to note that environmental effects were not included, in particular, drift effects due to temperature. This work focused on the calibration and traceability aspects, which are typical of laboratory calibration. The analysis of long-term stability, temperature, and humidity effect was out of the focus of this paper. It aimed to highlight the best way to use calibration data obtained at laboratory temperature and relative humidity, 20 ± 5 °C and 50 ± 10%, respectively. Furthermore, drift and environmental effects strongly depend on specific use cases, which could be different from each other.

Future work will focus on the long-term characterization of accelerometers under different working conditions using a climatic chamber. In particular, new MEMS designs aimed at minimizing environmental effects on the performance of these transducers will be analyzed, simulating controlled temperature and humidity conditions—even under critical ranges. These extreme conditions are essential for applications such as aerospace and industrial environments, where the use of these transducers is growing, making the assurance of their reliability a crucial task.

## Figures and Tables

**Figure 1 sensors-25-02120-f001:**
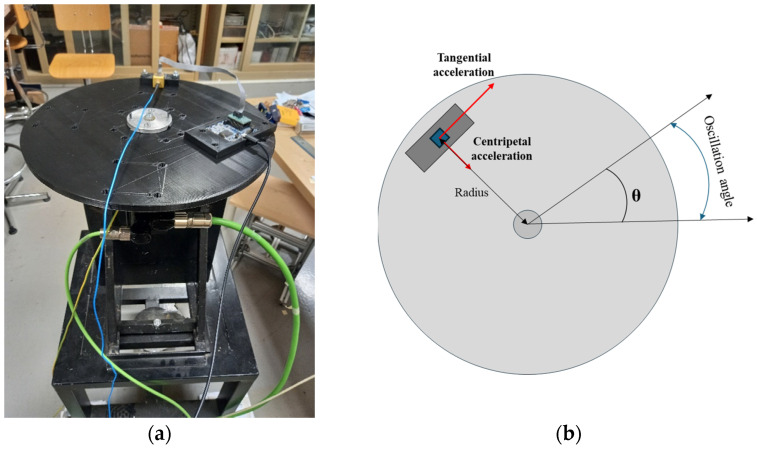
Picture of the calibration bench (**a**) and the scheme representing the rotation test bench functioning (**b**).

**Figure 2 sensors-25-02120-f002:**
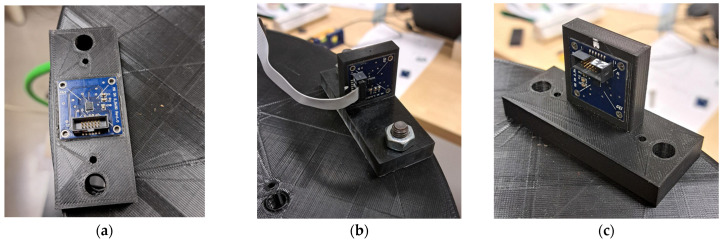
Pictures of the support used to orient the axes of micro-electromechanical systems (MEMS) in different positions with respect to the tangential and centripetal accelerations. (**a**) *z*-axis in the vertical direction; (**b**) *z*-axis in the tangential direction; (**c**) *z*-axis in the radial direction.

**Figure 3 sensors-25-02120-f003:**
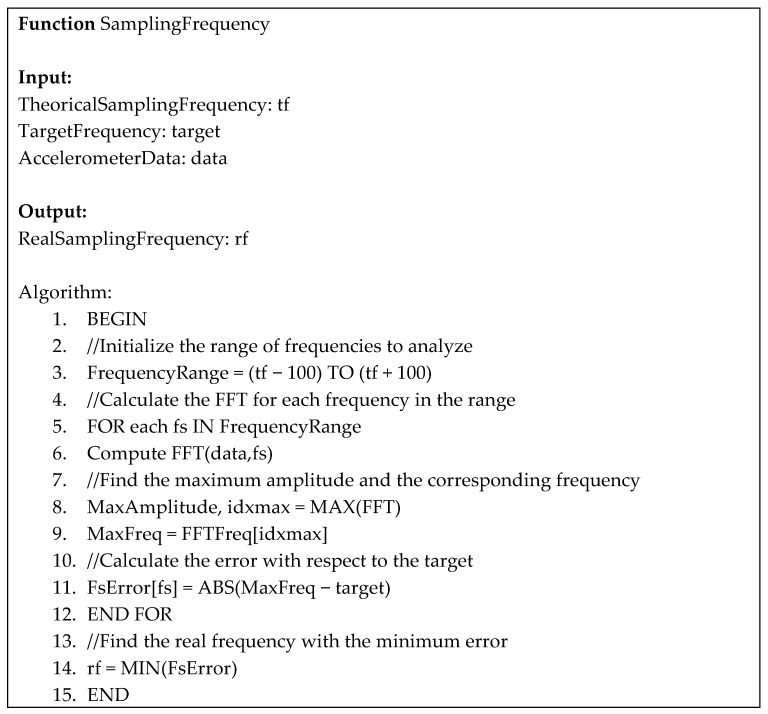
Structured algorithm description for the compensation of the sampling frequency variability of digital accelerometers.

**Figure 4 sensors-25-02120-f004:**
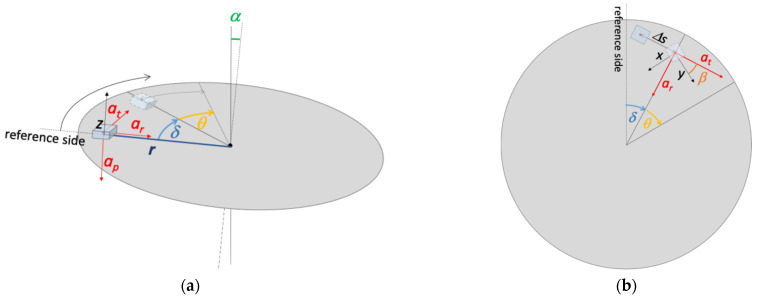
Scheme of the test bench: (**a**) lateral view; (**b**) top view. The indication of the *x*-*y*-*z* axes refers to the measuring reference system.

**Figure 5 sensors-25-02120-f005:**
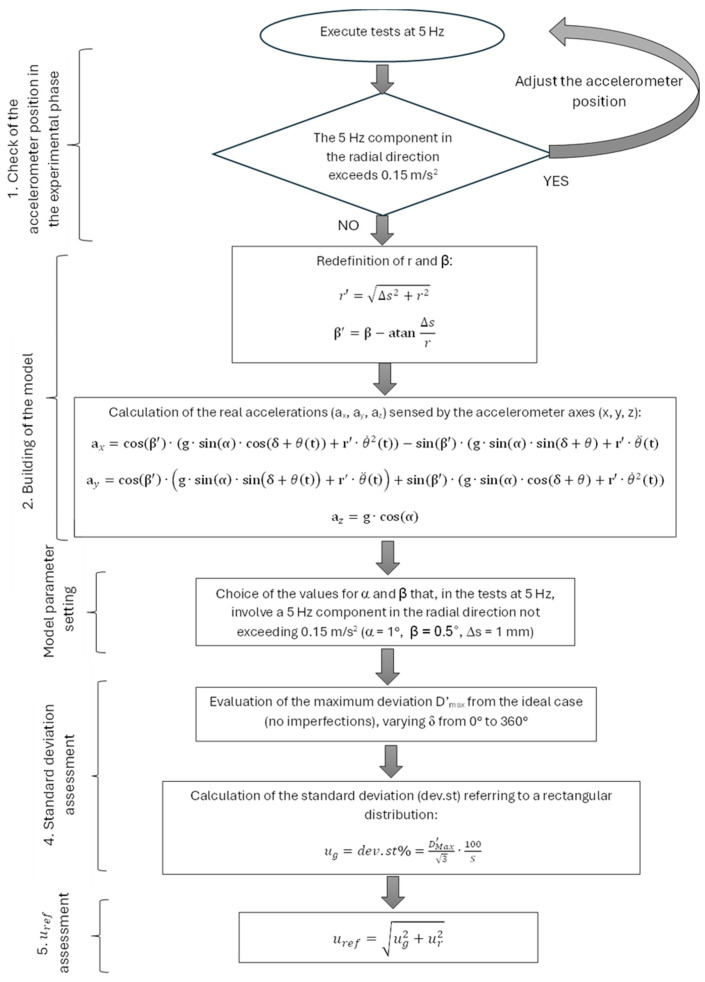
Diagram of the procedure for the evaluation of the uncertainty contribution due to geometrical imperfections of the bench.

**Figure 6 sensors-25-02120-f006:**
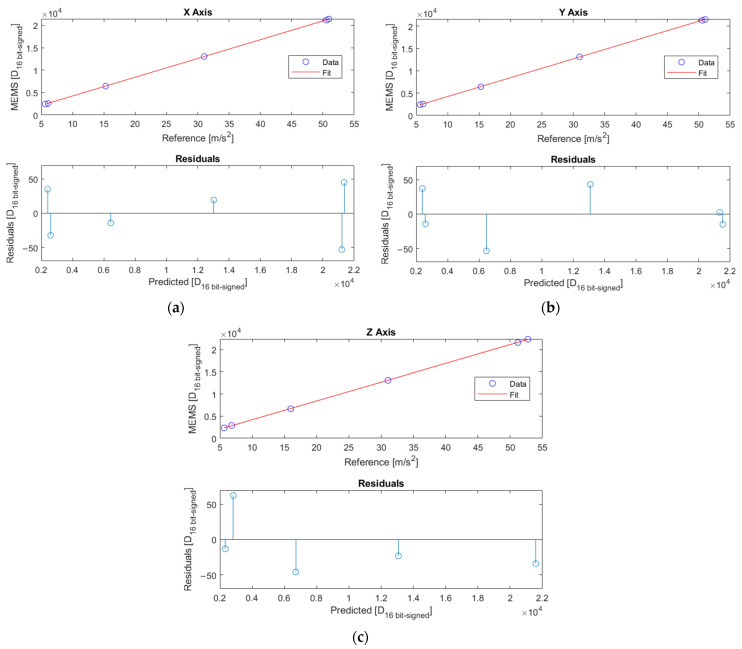
Linear interpolations and residual plots of the *x*-axis (**a**), *y*-axis (**b**), and *z*-axis (**c**) of one of the MEMS accelerometers. The reference comes from the encoder signal.

**Figure 7 sensors-25-02120-f007:**
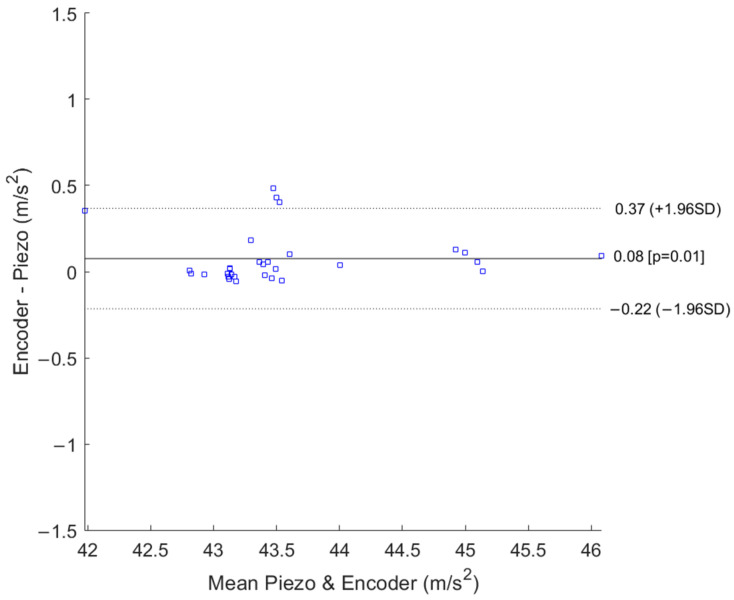
Bland–Altman plot for the tangential axis of the piezo IEPE accelerometer and the acceleration data provided by the encoder.

**Figure 8 sensors-25-02120-f008:**
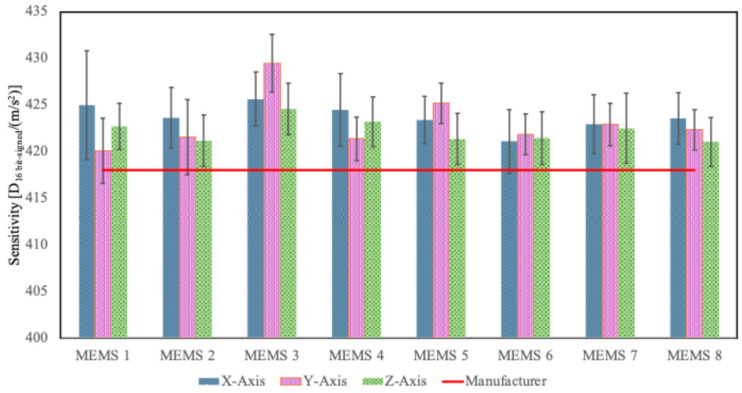
Evaluated sensitivity values for each axis of each accelerometer.

**Figure 9 sensors-25-02120-f009:**
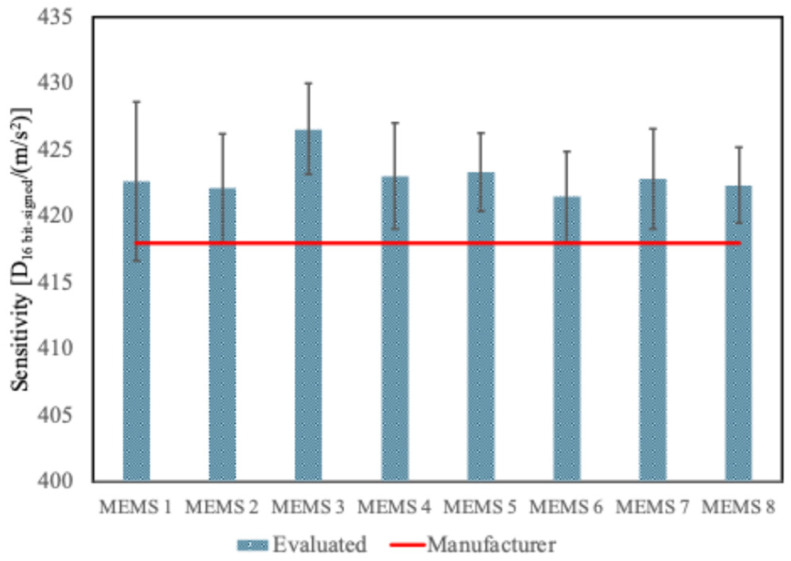
Evaluated sensitivities of all the studied MEMS considering a single value for each accelerometer.

**Figure 10 sensors-25-02120-f010:**
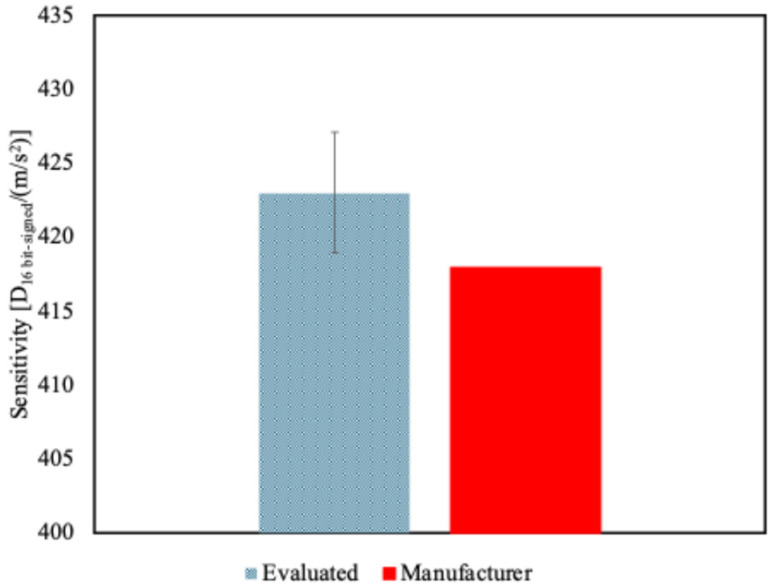
Evaluated single sensitivity value for all the MEMS independently of the axis or sensor compared to the manufacturer-provided values.

**Table 1 sensors-25-02120-t001:** Description of the model parameters.

Parameter	Description
α	Angle of inclination of the axis of rotation with respect to the vertical
β	Rotation angle of the accelerometer on the plane of the disk around its axis
Δs	Tangential displacement of the accelerometer
r	Distance of the sensitive element from the rotation axis
a_t_	Tangential acceleration at the sensitive element position
a_r_	Radial acceleration at the sensitive element position
a_p_	Acceleration perpendicular to the disk surface (gravity component)
δ	Initial angular position of the accelerometer
θ	Angular position assumed by the accelerometer during oscillations, from the initial angular position δ

**Table 2 sensors-25-02120-t002:** Standard uncertainty budget.

Type of Contribution	Symbol	Estimation
Repeatability	$u_{r}$	0.05%
Linearity	$u_{l}$	0.05%
Reference	$u_{ref}$	0.3%
**Calibration uncertainty**	$u_{c}=\sqrt{u_{r}^{2}+u_{l}^{2}+u_{ref}^{2}+2\cdot u_{r}\cdot u_{l}+2\cdot u_{r}\cdot u_{ref}+2\cdot u_{l}\cdot u_{ref}}=0.4\%$	

**Table 3 sensors-25-02120-t003:** Uncertainty estimates for the sensitivity calculated on the basis of groups of data.

Case	Group of Data	Overall Uncertainty
1	Single axis, single accelerometer	*x*: 0.8% *y*: 0.6% *z*: 0.7%
2	Single accelerometer	0.9%
3	All accelerometers	1%

## Data Availability

The original contributions presented in the study are included in the article, and further inquiries can be directed to the corresponding authors.

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
