# Peer review of "Dynamic Multi-Axis Calibration of MEMS Accelerometers for Sensitivity and Linearity Assessment"

_sensors, 2025, doi:10.3390/s25072120_

Round 1
Reviewer 1 Report (Previous Reviewer 4)
Comments and Suggestions for Authors
This study proposes a dynamic multi-axis calibration method for MEMS accelerometers based on a rotating test bench, where the sensitivity and linearity of the sensors are experimentally evaluated and an uncertainty budget is established. The research content has practical engineering significance, and it is suggested to be further revised and published in the journal.
These comments should be further revised:
1. The measurement method and traceability of the geometric parameters involved in the calculation of the reference acceleration, such as the tilt Angle α of the rotation axis and the displacement Δs, should be clarified.
2. The geometric error model assumes α≤1° and Δs≤1 mm, but these parameters' actual measurements or statistical distributions are not stated. Supplementary experimental data support is suggested.
3. Is the combination of the radius r (0.3%) uncertainty and the geometric error (0.03%) reasonable in the reference uncertainty calculation? Further explanation is needed.
Author Response
Please see the attachment.

Reviewer 2 Report (Previous Reviewer 1)
Comments and Suggestions for Authors
the authors have addressed the past comments
Author Response
Please see the attachment.

Reviewer 3 Report (New Reviewer)
Comments and Suggestions for Authors
This paper is devoted to the determining of the sensitivity and linearity of MEMS accelerometers. An extensive introduction describes the state of the art and existing approaches to solution of the problem. The method proposed in the paper allows one to generate periodic alternating accelerations up to 49 m/s^2 with good accuracy, which allows one to test the selected accelerometers. The results obtained in the paper allow one to determine the repeatability, linearity, and sensitivity of the devices under study. The paper can be published after several minor deficiencies are eliminated.
1. (Page 4, lines 150 and 155); the calculations are unclear. As far as I can understand, the accelerometer under study can measure acceleration in the range of [-8..+8]g and linearly represents it as a 16-bit signed number in the range of [-32768..32767]. This gives us an LSB value of 8/32767 = 0.244*10^-3*g and a sensitivity of 32767/(8*9.81) = 417.5 D per (m/s^2). The meanings of numbers 2161 and 21611 are unclear to me.
2. (Page 12, line 398 and futher); A reference is given to Section 2.4.2, which is missing from the text of the article.
3. (Figure 7, Page 13, line 427), the acceleration unit is mistakenly indicated as (m/s^-2).
4. The text of the article does not contain a reference to work 8.
Author Response
Please see the attachment.

This manuscript is a resubmission of an earlier submission. The following is a list of the peer review reports and author responses from that submission.
Round 1
Reviewer 1 Report
Comments and Suggestions for Authors
It is not clear if the presented work is an autonomous calibration, a non-autonomous or a combination of both.
Apparently, the authors conclude that the measured sensitivity of commercial accelerometers is better than the nominal one, which is not unexpected. It can hardly be a result from a research.
Also, the bench imperfections play a role on the proposed analysis, leading to consider that precision equipment should be used. It is not clear how the presented work is an improvement.
The discussion section and the conclusions are practically the same.
It is not clear how the proposed work would constitute calibrations, as the tools used in it are themselves inaccurate, and how the gravity field is used as a reference.
Comments on the Quality of English LanguageIt needs a major proofreading, there are grammar errors that should be adressed. Also, there are sentences which seem hard to understand. The quality of writing must be improved to convey the correct meaning of the presented work.
Reviewer 2 Report
Comments and Suggestions for Authors
This paper proposes a method for calibrating the sensitivity and linearity of an autonomous accelerometer and validates it through a commercial three-axis accelerometer. The experimental results also meet expectations.
I have a few questions for the author:
- In the measurement sampling range (as shown in Figure 5), why wasn't the 20-50 m/sec² range sampled? Would this have resulted in significant errors in linearity and repeatability in the measurements?
- Why do Figures 6 and 7 show larger error bars for MEMS1? Was this caused by the measurement process, or is it an inherent error in the sample itself? What is the reason for this?
- It would be even better if the measurement characteristics could be compared with those using a shaker vibration.
Reviewer 3 Report
Comments and Suggestions for Authors
This study presents a dynamic multi-axis calibration method for MEMS accelerometers using a custom-designed rotating test bench. The authors evaluated the sensitivity, linearity, and uncertainty of commercial triaxial MEMS accelerometers under varying acceleration conditions. The work highlights a comprehensive uncertainty budget and explores the feasibility of adopting a unified sensitivity value for batch-calibrated sensors. The experimental setup, data processing methods, and results are well-documented, offering potential value for structural health monitoring (SHM) and industrial applications. However, the following issues should be addressed to enhance the manuscript’s rigor and clarity.
1. Technical Issues
1.1 Calibration Methodology and Literature Context
- Insufficient Comparison with Existing Calibration Methods:
While the rotating bench approach is clearly described, the manuscript lacks a critical comparison with state-of-the-art autonomous and non-autonomous calibration methods (e.g., gravity-based or turntable-free techniques). Recent advancements in MEMS calibration, such as IEEE Sensors Journal, 2023, 23(12): 13319–13326; Micromachines, 2022, 13: 879, should be referenced to contextualize the novelty of the proposed method. - Statistical Significance of Results:
The conclusion that "differences among accelerometers and axes are not significant" (Page 15) requires statistical validation (e.g., ANOVA or confidence intervals). The current analysis relies on visual inspection of sensitivity plots (Fig. 6–8), which is insufficient. - Linearity Assessment:
The linearity evaluation (Section 3.1.2) lacks clarity. The manuscript states that linearity uncertainty is <0.05% but does not provide raw data or residual plots (Fig. 5 is overly simplified). A quantitative metric (e.g., R² values) should be included.
4. Additional Reference
The following literatures can be referenced, such as IEEE Sensors Journal, 2023, 23(12): 13319–13326; Micromachines, 2022, 13: 879; Journal of Microelectromechanical Systems, 2020, 29(1): 3; Sensors and Actuators A: Physical, 2022, 333: 113236.
1.2 Experimental Design and Reproducibility
- Sampling Frequency Variability:
The method for compensating sampling frequency variability (Page 6) is innovative but inadequately explained. A flowchart or pseudocode in the supplementary material would improve reproducibility. - Validation with External References:
The IEPE accelerometer is used for validation, but its alignment with MEMS data is only briefly mentioned (Page 11). A correlation plot or Bland-Altman analysis would better demonstrate consistency.
2. Other Issues
2.1 Language and Clarity
- Redundant Sections:
The Discussion section (Page 14–15) repeats content from Results (Page 12–13). Condense overlapping paragraphs to avoid redundancy. - Ambiguous Terminology:
- “Sensitivity” is interchangeably used for slope (Section 2.5.1) and calibration factor (Section 3.2). Clarify definitions.
- Abbreviations like “IEPE” and “PLA” are defined late in the text (Page 4 and Abbreviations). Define all abbreviations at first mention.
2.2 Typographical Errors
- Terminology:
- “young's modulus” → “Young’s modulus” (Page 2).
- “grups” → “groups”? (Section 3.3, Page 14).
3. Additional Recommendations
- Long-Term Stability Testing: Include data on sensor drift over extended periods to assess practical usability.
- Environmental Robustness: Evaluate temperature or humidity effects, even briefly, to address real-world applicability.
- Open Data: Provide raw data or code in a public repository to facilitate replication.
Key Strengths
- Detailed uncertainty modeling and experimental validation.
- Practical insights into batch calibration feasibility.
- Clear methodology for multi-axis dynamic testing.
Reviewer 4 Report
Comments and Suggestions for Authors
This paper presents a dynamic multi-axis calibration set to measure the sensitivity and linearity of MEMS accelerometers. It is unfortunate that its effectiveness needs further verification to prove its validity and therefore cannot be accepted. The reasons are as follows:
1、The paper mentions that geometric imperfections of the test bench (such as the tilt of the rotating axis, positioning deviation of the accelerometer, etc.) can affect the calculation of the reference acceleration. Although the author evaluated these impacts through a model, these imperfections themselves may introduce significant uncertainties. For example, the maximum geometric deviation mentioned in the article is 1.2%. Although it may be acceptable in some cases, it could still be non-negligible for high-precision applications.
2、The paper does not compare the proposed calibration set and method with other known calibration methods. For example, compared with traditional non-autonomous calibration methods (such as using a high-precision turntable). Without such a comparison, it is difficult to judge the practical application value of this method. If the test accuracy is not up to a comparable level, the low cost can not reflect its advantages, and can not be further applied.
3、The paper simply combines the repeatability, linearity, and reference uncertainties to obtain the total uncertainty, but does not discuss in detail the correlations among these uncertainty sources. In practical applications, there may be interactions among these uncertainty sources, and simply adding them together may underestimate or overestimate the total uncertainty.
4、Although the paper proposes a low-cost calibration method, in practical applications, this simple test bench may make it difficult to achieve high-precision calibration. For example, in applications that require high-precision measurements, such as aerospace and precision engineering, this calibration method may not meet the requirements.
5、The linearity evaluation mentioned in the paper is based on the output data at different acceleration amplitudes. However, the tested acceleration range (from 4 m/s² to 50 m/s²) may be insufficient to comprehensively assess the linearity of the accelerometer across its entire operating range. Especially for MEMS accelerometers, their non-linear characteristics may be more pronounced at higher or lower accelerations.
6、More importantly for the test range, the points tested were mainly centered between approximately less than 10 m/s² and 50-55 m/s² (Figure 5), which hardly suggests that the device can its ability to test over the entire interval, and limits its application value.
7、The reference acceleration is calculated based on encoder signals and a theoretical model rather than being directly measured. This indirect method may introduce errors due to inaccurate model assumptions (such as assuming perfect sinusoidal motion). In addition, the uncertainties of the reference acceleration mentioned in the article mainly come from the accuracy of the encoder and the geometric imperfections of the test bench, but other possible error sources (such as the dynamic characteristics of the motor, vibration interference, etc.) are not discussed in detail.